# Unsupervised Optical Flow Estimation with Dynamic Timing Representation for Spike Camera

**Lujie Xia**[1,2], **Ziluo Ding**[2,3], **Rui Zhao**[1,2], **Jiyuan Zhang**[1,2],
**Lei Ma**[5], **Zhaofei Yu**[1,2,4], **Tiejun Huang**[1,2], **Ruiqin Xiong**[1,2*]

[1]School of Computer Science, Peking University
[2]National Engineering Research Center of Visual Technology (NERCVT)
[3]Beijing Academy of Artificial Intelligence
[4]Institute for Artificial Intelligence, Peking University
[5]National Biomedical Imaging Center, College of Future Technology, Peking University
{lujie.xia, ziluo, lei.ma, yuzf12, tjhuang, rqxiong}@pku.edu.cn
{ruizhao, jyzhang}@stu.pku.edu.cn

## Abstract

Efficiently selecting an appropriate spike stream data length to extract precise information is the key to the spike vision tasks. To address this issue, we propose a dynamic timing representation for spike streams. Based on multi-layers architecture, it applies dilated convolutions on temporal dimension to extract features on multi-temporal scales with few parameters. And we design layer attention to dynamically fuse these features. Moreover, we propose an unsupervised learning method for optical flow estimation in a spike-based manner to break the dependence on labeled data. In addition, to verify the robustness, we also build a spike-based synthetic validation dataset for extreme scenarios in autonomous driving, denoted as SSES dataset. It consists of various corner cases. Experiments show that our method can predict optical flow from spike streams in different high-speed scenes, including real scenes. For instance, our method achieves $15\%$ and $19\%$ error reduction on PHM dataset compared to the best spike-based work, SCFlow, in $\Delta t = 10$ and $\Delta t = 20$ respectively, using the same settings as in previous works. The source code and dataset are available at https://github.com/Bosserhead/USFlow.

## 1 Introduction

Optical flow is defined as the apparent motion of individual pixels on the image plane and is prevalent for being an auxiliary tool for various vision tasks, *e.g.* frame rate conversion [15], scene segmentation [43, 38], and object detection [49]. In high-speed scenes, the optical flow estimation may suffer from blurry images from low frame-rate traditional cameras. Obtaining the data with a device that can precisely record the continuous light intensity changes of the scene is the key to addressing this issue. Recently, neuromorphic cameras have been developed greatly, such as event camera and spike camera. They can record light intensity changes in high-speed scenes. Especially, for spike camera, each pixel of it responds independently to the accumulation of photons by generating asynchronous spikes. It records full visual details with an ultra-high temporal resolution (up to 40kHZ). With these features, spike camera has demonstrated superiority in handling some high-speed scenarios, such as video frame interpolation [35] and dynamic scene reconstruction [40, 42, 7, 8, 48].

Since spike camera can record details of high-speed moving objects, it has enormous potential for estimating more accurate optical flow in high-speed scenes. Considering that deep learning has

---

*Corresponding author.

achieved remarkable success in frame-based optical flow estimation [31, 32], it seems reasonable to directly apply frame-based architecture in spike data. However, the data modality of spike stream output by spike camera is quite different from frame images. For each pixel in spike camera, a spike is fired and the accumulation is reset when photons accumulation at a pixel exceeds a set threshold. At each timestamp, the spike camera outputs a binary matrix, denoted as spike frame, representing the presence of spikes at all pixels. Previous work [12] utilizes spike stream as naïve input representation for one timestamp, which consists of a series of spike frames within a predefined time window. However, a too-long window can incur lots of misleading frames, while a short window is sensitive to noise and cannot provide enough information. Therefore, deliberate modifications are needed for input representation to extract salient information more flexibly and efficiently from spike streams before optical flow estimation architecture takes over.

In addition, the ground truth of optical flow is scarce in the real world, especially for high-speed scenes. To cope with the lack of labeled real-world data, it is necessary to study spike-based optical flow estimation in an unsupervised manner. As described above, the light intensity information is contained within the spike intervals. This difference in data characteristics makes it unreasonable to directly apply the frame-based unsupervised loss to spike streams. Therefore, the light intensity should first be extracted from spike streams in the spike-based unsupervised loss. This is also the core of constructing the illuminance consistency on spike streams. Moreover, we argue the field of autonomous driving is a good place to validate spike camera since it is suitable for high-speed scenes. In autonomous driving, it is nearly impossible to collect real data for complex, diverse, high-speed extreme scenarios, *e.g.* vehicle collisions and pedestrian-vehicle accidents. However, these scenarios are of great significance to improve the safety of this field and should be highlighted. Therefore, in order to verify that the spike-based algorithm can handle extreme scenarios, we propose a spike-based synthetic validation dataset for extreme scenarios in autonomous driving, denoted as the SSES dataset.

In this paper, we propose an unsupervised method for spike-based optical flow estimation with dynamic timing representation, named USFlow. In our unsupervised loss, we propose two strategies, multi-interval-based and multi-time-window-based, to estimate light intensity in regions with different motion speeds. The estimated optical flow is utilized to distinguish regions with different motion speeds and generates corresponding weights for light intensity fusion. Then the final approximate light intensity participates in loss calculation. As for addressing the fixed time window issue, there is a way that apply a dynamic time window to the different spike streams. To this end, we propose Temporal Multi-dilated Representation(TMR) for spike streams. In more detail, we apply multi-layer dilated convolutions to operate on the temporal dimension of spike streams. Multi-layer dilated convolutions enable the network to have different receptive fields and each layer can be regarded as summarizing spike stream with one different time window. We also design a Layer Attention(LA) module to extract salient features and filter the redundant ones.

Following the settings in previous works [12], we train our method on SPIFT [12] dataset and evaluate it on PHM [12] and our proposed SSES datasets. We demonstrate its superior generalization ability in different scenarios. Results show that USFlow outperforms all the existing state-of-the-art methods qualitatively and quantitatively. USFlow shows visually impressive performance on real-world data.

## 2 Related Works

**Deep Learning in Optical Flow Estimation.** Frame-based optical flow estimation is a classical task in the computer vision area through the years and has been solved well. PWC-Net[31] and Liteflownet[13] introduce the pyramid and cost volume to neural networks for optical flow, warping the features in different levels of the pyramid and learning the flow fields in a coarse-to-fine manner. RAFT[32] utilizes ConvGRU to construct decoders in the network and iteratively decodes the correlation and context information in a fixed resolution. Due to their excellent performance, PWC-Net and RAFT are the backbones of most algorithms[37, 34, 41, 21, 20, 18, 17, 36, 16, 39, 30] in frame-based optical flow estimation. TransFlow [22] utilizing the long-range temporal association to recover more information by introducing transformer architecture. In addition, many frame-based unsupervised optical flow networks[25, 21, 20, 18, 23, 30] were proposed to discard the need for labeled data. Similar to the traditional optimization-based methods, Yu *et al.*[14] employ photometric loss and smoothness loss to train a flow estimation network. Unflow[25] applies a forward-backward

check on bidirectional optical flow to estimate the occlusion area, where the backpropagation of photometric loss is stopped.

Deep learning has also been applied to event-based optical flow[45, 46, 19, 6, 11, 26, 5]. EV-FlowNet[45] can be regarded as the first deep learning work training on large datasets with a U-Net architecture[28]. As an updated version of EV-FlowNet, an unsupervised framework has been proposed by Zhu *et al.*[46] based on contrast maximization. Spike-FlowNet[19] and STE-FlowNet[6] use spiking neural networks and ConvGRU to extract the spatial-temporal features of events, respectively.

The research on spike-based optical flow estimation is just getting started. SCFlow[12], is the first deep-learning method. It proposes a large dataset, the SPIFT dataset, to train an end-to-end neural network via supervised learning. However, our work aims to fill in the blanks of unsupervised learning.

**Event-based and Spike-based Input Representation.** Normally, asynchronous event streams are not compatible well with the frame-based deep learning architecture. Therefore, frame-like input representation is needed and is expected to capture rich salient information about the input streams. Apart from many handcrafted representations[46, 27, 24, 29, 6], some end-to-end learned representations have been proposed, making it possible to generalize to different vision tasks better. Gehring *et al.*[10] simply uses multi-layer perceptrons as a trilinear filter to produce a voxel grid of temporal features. Cannici *et al.*[4] propose Matrix-LSTM, a grid of Long Short-Term Memory (LSTM) cells that efficiently process events and learn end-to-end task-dependent event surfaces. Similarly, Event-LSTM [1] utilizes LSTM cells to process the sequence of events at each pixel considering it into a single output vector that populates that 2D grid. Moreover, Vemprala *et al.*[33] presents an event variational autoencoder and shows that it is feasible to learn compact representations directly from asynchronous spatio-temporal event data.

Directly using spike stream as input might incur lots of misleading information or miss something necessary if the time window is not chosen appropriately. Therefore, frame-like input representation should also be carefully designed to extract sufficient information from spike stream. SCFlow [12] uses estimated optical flow to align spike frames to eliminate motion blur.

Different from all previous works, we aim to train an end-to-end input representation with the function of a dynamic time window by multi-layer dilated convolutions.

## 3 Preliminaries

### 3.1 Spike Camera

Spike camera works by an "integrate-and-fire" mechanism, which asynchronously accumulates the light on each pixel. The integrator of spike cameras accumulates the electrons transferred from incoming photons. Once the cumulative electrons exceed a set threshold, the camera fires a spike and resets the accumulation. The process of accumulation can be formulated as,

$$\mathbf{A}(\mathbf{x}, t) = \int_0^t \alpha I(\mathbf{x}, \tau) d\tau \mod \theta, \tag{1}$$

where $\mathbf{A}(\mathbf{x}, t)$ is the cumulative electrons at pixel $\mathbf{x} = (x, y)$. $I(\mathbf{x}, \tau)$ is the light intensity at pixel $\mathbf{x}$ at time $\tau$. $\alpha$ is the photoelectric conversion rate. $\theta$ is the threshold. The reading time of spikes is quantified with a period $\delta$ of microseconds. The spike camera fires spikes at time $T$, $T = n\delta, n \in \mathbf{Z}$, and generate an $H \times W$ spike frame $s$. As time goes on, the camera produces a spatial-temporal binary stream $S_t^N$ in $H \times W \times N$ size, as shown in Figure 2. The $N$ is the temporal length of the spike stream. $H$ and $W$ are the height and width of the sensor, respectively.

### 3.2 Problem Statement

Given two timestamps $t_0$ and $t_1$, we have two spike streams centered on $t_0$ and $t_1$, noted as $S_{t_0}^L$ and $S_{t_1}^L$, respectively. Then we estimate a dense displacement field $\mathbf{f} = (f^u, f^v)$ from $t_0$ to $t_1$ using

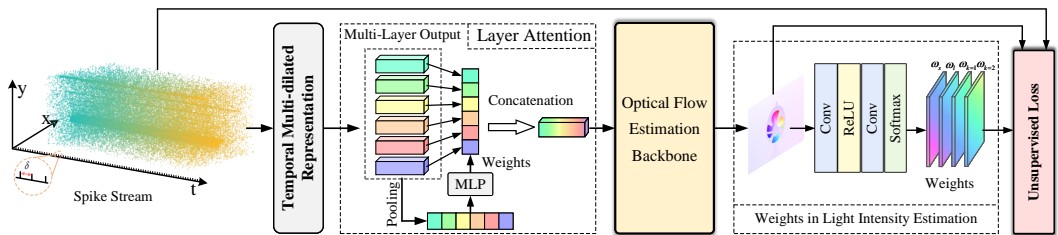

Figure 1: The overall architecture of the USFlow. First, The input spike stream is through the temporal multi-dilated representation, and the layer attention fuses the multi-layer outputs. Second, the fused output is fed to the optical flow estimation backbone. Finally, it uses estimated optical flow to learn weights for light intensity estimation.

these two sub-spike streams, mapping each pixel $\mathbf{x} = (x, y)$ at $t_0$ to its corresponding coordinates $\mathbf{x}' = (x', y') = (x + f^u(\mathbf{x}), y + f^v(\mathbf{x}))$ at $t_1$.

# 4 Method

## 4.1 Overview

To verify the effectiveness of the proposed dynamic timing representation, we present two versions of USFlow. One is the PWC-like version. It adopts a variant of PWC-Net [31] which is also the backbone of SCFlow [12]. The other is the RAFT version which adopts an official small version of RAFT [32] as the backbone. More about these two backbones is included in the appendix. Considering one advantage of neuromorphic vision is low latency, our representation part should be lightweight and efficient. The two spike streams first pass into the shared dynamic timing representation module separately, whose outputs are sent into existing backbones. In addition, we design an unsupervised loss to break the dependency on labeled data. Note that all the components are trained end-to-end.

## 4.2 Dynamic Timing Representation

As stated in 3.1, one binary spike frame itself is sensitive to noise and meaningless without contextual connection. Given a bunch of spike data, selecting an appropriate data length is pivotal for subsequent processing. A too-long spike stream is not suitable for high-speed regions since time offset accumulation introduces more redundant information. A too-short spike stream won't work either, as it can not exhibit light intensity precisely with few binary data. To address this issue, we propose a dynamic timing representation for input spike streams.

The Dynamic Timing Representation consists of Temporal Multi-dilated Representation (TMR) module and a Layer Attention module (LA). The main ingredient of TMR is dilated convolution. By using dilated convolution, we can extend receptive fields to a larger range with just a few stacked layers, while preserving the input resolution throughout the network as well as computational efficiency. In more detail, we apply 1D dilated convolutions to the spike stream for each pixel and the parameters are shared across all the pixels. The TMR can be formulated as follow,

$$\{F^{(i)}(\mathbf{x})\} = \mathrm{D1C}^{(i)}(F^{(i-1)}(\mathbf{x})) \quad i = 1, \ldots, n, \tag{2}$$

where $\mathrm{D1C}(\cdot)$ represents the 1D dilated convolution operation, $i$ and $\mathbf{x}$ are layer index and spatial coordinate respectively. Note that $F^{(0)}(\mathbf{x})$ is the input spike stream on position $\mathbf{x}$. Figure 2 depicts dilated convolutions for dilations 1, 2, 4, and 8. The higher level of the layer, the larger the receptive field is. The intuition behind this configuration is two-fold. First, a different layer can be regarded as summarizing spikes or extracting temporal correlation among spike frames with a different time window. Second, dilated convolution allows the network to operate on a coarser scale more effectively than with a normal convolution, which guarantees the model is lightweight. In Table 1, we show the parameter size of our USFlow and other methods.

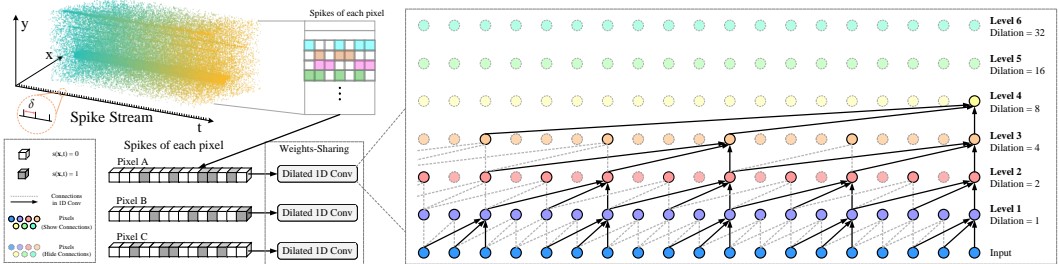

Figure 2: The Temporal Multi-dilated Representation. 1D dilated convolutions are applied to all pixels with shared weights. The dilated convolutions are stacked for larger receptive fields. The higher level the layer is, the larger the receptive field is. Note that the illustration of the spike stream is shown in the top-left corner.

Multi-dilated architecture has already greatly expanded the input dimensions but not all layers provide equally useful information. Blindly fusing the output of all layers may impair the learning process. To address this issue, we propose the Layer Attention module to further flexibly emphasize which layer is meaningful or which time window is optimal. As illustrated in Figure 1, we average the output of $n$ layers, $\{F^{(i)}(\mathbf{x})\}_{i=1}^{n}$, noted as $F'(\mathbf{x})$ to generate one $n$-dimension layer context descriptor. The descriptor is then forwarded to a multi-layer perceptron(MLP) to produce our layer attention map. The layer attention values are broadcast along the layer dimension and the final concatenation, $RF(\mathbf{x})$, is computed as follows:

$$RF(\mathbf{x}) = \sigma(\mathrm{MLP}(\mathrm{AvgPool}(\mathbf{F}'(\mathbf{x})))) \otimes \mathbf{F}'(\mathbf{x}), \tag{3}$$

here $\otimes$ denotes element-wise multiplication. We do this operation on all pixels by sharing weights.

### 4.3 Unsupervised Loss

Unsupervised learning is necessary for spike-based optical flow estimation due to the lack of labeled training data in high-speed real-world scenes. To this end, we sidestep the need for ground truth of optical flow as only spike data are required for training. More specifically, at training time, two spike streams centered on $t_0$ and $t_1$, $S_{t_0}^{L}$ and $S_{t_1}^{L}$, are utilized to predict optical flow, $\mathbf{f} = (f^u, f^v)$, from $t_0$ to $t_1$.

Different from traditional RGB frames, a binary spike frame can not accurately represent the light intensity at the current timestamp. Therefore, estimating precise current light intensity from binary spike streams is the key to constructing an unsupervised loss function.

**For low-speed regions.** Since the light intensity accumulation in low-speed regions is an approximately linear process, we count spikes to utilize longer-duration information to estimate the light intensity. In this way, we can improve the robustness of light intensity estimation. However, the data length of the spike stream used to estimate light intensity varies with different low-speed motions. For reducing computational complexity, we only set two different time windows. The light intensity estimation can be formulated as follow:

$$\tilde{I}_T(\mathbf{x}, \tau) = \frac{\omega_s \cdot \theta}{2D_s + 1} \cdot \sum_{t=\tau-D_s}^{\tau+D_s} s(\mathbf{x}, t) + \frac{\omega_l \cdot \theta}{2D_l + 1} \cdot \sum_{t=\tau-D_l}^{\tau+D_l} s(\mathbf{x}, t), \tag{4}$$

here $D_s$ and $D_l$ are the half length of time windows. They are set to 40 and 100 respectively. $\omega_s$ and $\omega_l$ are the weight factors. The subscripts $s$ and $l$ refer to short and long time windows, respectively.

**For high-speed regions.** Selecting a large time span of spike streams to extract information would incur motion blur due to time offset accumulation. Hence, we estimate light intensity during a single spike interval which is typically on the order of microseconds. Since the spike streams have extremely high temporal resolution and the interval between adjacent spikes is ultra-short, we can safely assume that the light intensity remains constant during the interval [47]. Let us denote $s(\mathbf{x}, m)$ and $s(\mathbf{x}, n)$

as two adjacent spikes at position $\mathbf{x}$. $m$ and $n$ are the timestamps corresponding to these two spikes. According to the spike camera working mechanism in Equation 1, the constant light intensity can be approximated by:

$$\hat{I}(\mathbf{x}) \approx \frac{\theta}{\alpha \cdot (n - m)} \quad , m < n \tag{5}$$

In reality, however, the number of incoming photons in an ultra-short interval is a random variable subject to the Poisson distribution even under constant illumination conditions. Therefore, the light intensity calculated by Equation 5 includes errors due to random fluctuations. To address this issue, we extend it by multi-intervals and fuse light intensity information with different numbers of intervals,

$$\tilde{I}_I(\mathbf{x}, \tau) = \sum_{k=1}^{K} (\omega_k \cdot \frac{(2k - 1) \cdot \theta}{\alpha \cdot [T(\mathbf{x}, N_\tau(\mathbf{x}) + k - 1) - T(\mathbf{x}, M_\tau(\mathbf{x}) - k + 1)]}), \tag{6}$$

$$\text{where} \quad M_\tau(\mathbf{x}) = \arg\max_z(T(\mathbf{x}, z) < \tau), \quad N_\tau(\mathbf{x}) = \arg\min_z(T(\mathbf{x}, z) \geq \tau). \tag{7}$$

In Equation 6 and 7, $T(\mathbf{x}, z)$ refers to the timestamp corresponding to the $z$th spike at position $\mathbf{x}$. $[T(\mathbf{x}, N_\tau(\mathbf{x}) + k - 1) - T(\mathbf{x}, M_\tau(\mathbf{x}) - k + 1)]$ is the total time length of these $(2k - 1)$ intervals. $\omega_k$ is the weight factor of the light intensity calculated by using $(2k - 1)$ intervals. Note that $k$ is set to 1 and 2 in our experiments. This setting ensures that the data length of spike stream used for light intensity estimation in high-speed motion regions is much shorter than that used in low-speed regions. Further detailed discussion is presented in the appendix.

**Learnable weights of estimated light intensity.** Considering a scene may contain both high-speed regions and low-speed regions, it is necessary for our unsupervised loss to fuse light intensity estimation methods based on multiple intervals and multiple time windows. We use the estimated optical flow to reflect the motion speed, and choose the most appropriate light intensity estimation strategy for different regions. As shown in Figure 1, we learn the weights $\omega_s$, $\omega_l$, $\omega_k$ from the estimated optical flow. Thence, we can fuse all the terms in Equation 4 and Equation 6 to achieve the final approximated light intensity $\tilde{I}$.

After achieving the $\tilde{I}$, we then derive the bidirectional photometric loss in a spike-based manner:

$$\mathcal{L}_{\text{photo}}(\mathbf{f}, \mathbf{f}') = \sum_{\mathbf{x}} (\rho(\tilde{I}(\mathbf{x}, t_0) - \tilde{I}(\mathbf{x} + \mathbf{f}, t_1)) + \rho(\tilde{I}(\mathbf{x} + \mathbf{f}', t_0) - \tilde{I}(\mathbf{x}, t_1))), \tag{8}$$

where $\rho$ is the Charbonnier loss [3], $\mathbf{f}$ is the flow from $t_0$ to $t_1$, $\mathbf{f}'$ is the flow from $t_1$ to $t_0$.

Furthermore, we use smoothness loss to regularize the predicted flow. It minimizes the flow difference between neighboring pixels, thus it can enhance the spatial consistency of neighboring flows and mitigate some other issues, *e.g.* the aperture problem. It can be written as:

$$\mathcal{L}_{\text{smooth}}(f, f') = \frac{1}{HW} \sum_{\mathbf{x}} |\nabla f(\mathbf{x})| + |\nabla f'(\mathbf{x})|, \tag{9}$$

where $\nabla$ is the difference operator, $H$ is the height and $W$ is the width of the predicted flow. The total loss function consists of two loss terms above-mentioned, which can be written as $\mathcal{L}_{\text{total}} = \mathcal{L}_{\text{photo}} + \lambda \mathcal{L}_{\text{smooth}}$, where $\lambda$ is the weight factor.

## 5 Experiments

### 5.1 Implementation Details

In this work, we choose the SPIFT dataset [12] as the training dataset. The PHM dataset [12] and our proposed SSES dataset is used for evaluation. The SPIFT and PHM datasets provide two data settings, generating optical flow every 10 spike frames ($\Delta t = 10$) and 20 spike frames ($\Delta t = 20$) separately from the start to end of sampling. Therefore, we train models for the ($\Delta t = 10$) and

Table 1: Average end point error (AEE) comparison with other methods for estimating optical flow on PHM datasets under $\Delta t = 10$ and $\Delta t = 20$ settings. All methods use spike stream as input and are trained on SPIFT dataset *in a supervised manner*. The best results are marked in bold.

| | Method | Param. | Ball | Cook | Dice | Doll | Fan | Hand | Jump | Poker | Top | Mean |
|---|---|---|---|---|---|---|---|---|---|---|---|---|
| $\Delta t = 10$ | EV-FlowNet | 53.43M | 0.567 | 3.030 | 1.066 | 1.026 | 0.939 | 4.558 | 0.824 | 1.306 | 2.625 | 1.771 |
| | Spike-FlowNet | 13.04M | 0.500 | 3.541 | **0.666** | 0.860 | 0.932 | 4.886 | 0.878 | 0.967 | 2.624 | 1.762 |
| | SCFlow | 0.80M | 0.671 | 1.651 | 1.190 | 0.266 | 0.298 | 1.692 | 0.120 | 1.030 | 2.166 | 1.009 |
| | RAFT | 0.99M | 0.577 | 1.557 | 1.135 | 0.296 | 0.327 | 1.769 | 0.141 | 0.681 | 2.198 | 0.965 |
| | PWC-Net(variant) | 0.57M | 0.597 | 2.185 | 1.288 | 0.606 | 0.464 | 2.551 | 0.370 | 1.269 | 2.602 | 1.326 |
| | RAFT+conv | 1.04M | 0.535 | 1.874 | 1.225 | 0.737 | 0.447 | 2.482 | 0.299 | 0.957 | 2.424 | 1.220 |
| | PWC-Net(variant)+conv | 0.62M | 0.511 | 1.585 | 0.959 | **0.187** | 0.245 | 1.874 | **0.077** | 0.833 | 2.114 | 0.932 |
| | USFlow(raft) | 1.04M | 0.483 | **1.228** | 1.294 | 0.283 | 0.345 | 1.634 | 0.154 | 0.764 | 2.192 | 0.931 |
| | USFlow(pwc) | 0.62M | **0.430** | 1.556 | 0.787 | 0.210 | **0.226** | **1.615** | 0.105 | **0.646** | **2.111** | **0.854** |
| $\Delta t = 20$ | EV-FlowNet | 53.43M | 1.051 | 5.536 | 1.721 | 2.057 | 1.867 | 8.820 | 1.803 | 2.193 | 5.061 | 3.345 |
| | Spike-FlowNet | 13.04M | 0.923 | 7.069 | **1.131** | 1.675 | 1.838 | 9.829 | 1.701 | 1.373 | 5.257 | 3.422 |
| | SCFlow | 0.80M | 1.157 | 3.430 | 2.205 | 0.507 | 0.578 | 4.018 | 0.267 | 1.922 | 4.327 | 2.046 |
| | RAFT | 0.99M | 1.004 | 3.378 | 2.059 | 0.460 | 0.561 | 3.707 | 0.257 | 1.416 | 4.250 | 1.904 |
| | PWC-Net(variant) | 0.57M | 1.321 | 4.493 | 2.601 | 2.206 | 1.083 | 5.654 | 1.159 | 2.320 | 5.143 | 2.887 |
| | RAFT+conv | 1.04M | 0.983 | 2.977 | 1.864 | 0.533 | 0.622 | 3.421 | 0.287 | 1.361 | 4.313 | 1.818 |
| | PWC-Net(variant)+conv | 0.62M | 0.881 | 3.198 | 1.624 | 0.799 | 0.476 | 4.030 | 0.273 | 1.424 | 4.324 | 1.892 |
| | USFlow(raft) | 1.04M | **0.792** | **2.734** | 1.918 | 0.448 | 0.569 | **2.601** | 0.256 | 1.231 | 4.293 | **1.649** |
| | USFlow(pwc) | 0.62M | 0.807 | 3.075 | 1.613 | **0.370** | **0.377** | 3.663 | **0.168** | **1.216** | **4.216** | 1.723 |

Table 2: Average end point error (AEE) comparison with other methods for estimating optical flow on PHM datasets under $\Delta t = 10$ and $\Delta t = 20$ settings. All methods use spike stream as input and are trained on SPIFT dataset *in an unsupervised manner*. The best results are marked in bold.

| | Method | Ball | Cook | Dice | Doll | Fan | Hand | Jump | Poker | Top | Mean |
|---|---|---|---|---|---|---|---|---|---|---|---|
| $\Delta t = 10$ | PWC-Net(variant) | 0.642 | 3.239 | 1.258 | 0.579 | 0.844 | 4.799 | 0.587 | 1.468 | 2.632 | 1.783 |
| | USFlow(pwc, Census) | **0.640** | 3.549 | **0.675** | 0.815 | 1.018 | 4.942 | 0.782 | **0.756** | 2.655 | 1.759 |
| | USFlow(pwc, SSIM) | 0.703 | 3.546 | 0.705 | 0.838 | 0.980 | 5.000 | 0.762 | 0.819 | 2.644 | 1.777 |
| | USFlow(pwc) | 0.705 | **2.170** | 1.416 | **0.555** | **0.610** | **2.219** | **0.438** | 1.159 | **2.488** | **1.307** |
| $\Delta t = 20$ | PWC-Net(variant) | 1.480 | 6.682 | 1.419 | 1.407 | 1.452 | 9.307 | 0.905 | 1.790 | 5.580 | 3.336 |
| | USFlow(pwc, Census) | **1.117** | 7.049 | **1.199** | 1.513 | 1.858 | 9.854 | 1.441 | **1.393** | 5.284 | 3.412 |
| | USFlow(pwc, SSIM) | 1.152 | 6.882 | 1.273 | 1.071 | 1.260 | 9.778 | 3.185 | 1.554 | 4.855 | 3.446 |
| | USFlow(pwc) | 1.259 | **3.978** | 2.782 | **0.784** | **0.873** | **4.748** | **0.524** | 2.180 | **4.589** | **2.413** |

($\Delta t = 20$) settings separately as SCFlow [12]. All training details are included in the appendix. More details on the SSES dataset will be elaborated in Section 5.4. The color map used in visualization refers to Middlebury [2].

## 5.2 Comparison Results

**Evaluation of Input Representation.** To fully validate the effectiveness of the proposed input representation, we first train the model in a supervised manner and compare it with other supervised baselines listed in the prior spike-based work, SCFlow [12]. In more detail, apart from SCFlow, we compare our network with baselines in event-based optical flow, *i.e.* EV-FlowNet [44] and Spike-FlowNet [19]. We also compare our network with frame-based optical flow network, *i.e.* RAFT [32] and PWC-Net(variant) [31]. Note that these two frame-based networks are lightweight versions as illustrated in Section 4.1 and we implement our method on both networks, denoted as USFlow(raft) and USFlow(pwc). All the methods in Table 1 are only fed spike streams as inputs.

As illustrated in Table 1, our input representation indeed can improve the performance on top of frame-based backbones. It demonstrates the necessity of directly pre-processing operations on spike stream information. In addition, USFlow(pwc) achieves the best mean AEE of 0.854 in $\Delta t = 10$ setting and USFlow(raft) gets the best mean AEE of 1.649 in $\Delta t = 20$ setting, which gets **15%** and **19%** error reduction from the best prior deep network, SCFlow, respectively. Note that the mean AEE value is averaged over nine scenes. Meanwhile, our input representation has the least parameters compared to other methods. The parameter size of input representation in SCFlow is 0.23M and it in our USFlow is only **0.05M**. With the lightweight representation module, the computational time of USFlow for inferring optical flow between two timestamps on a 3090 GPU is 90.6ms. It gets 61.8% computational time reduction from SCFlow.

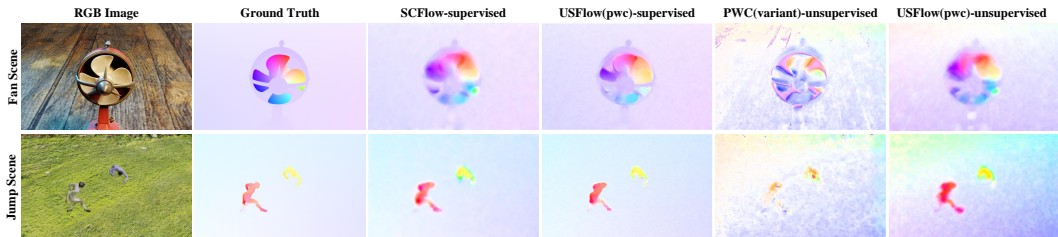

Figure 3: Qualitative results of the evaluation of the PHM dataset.

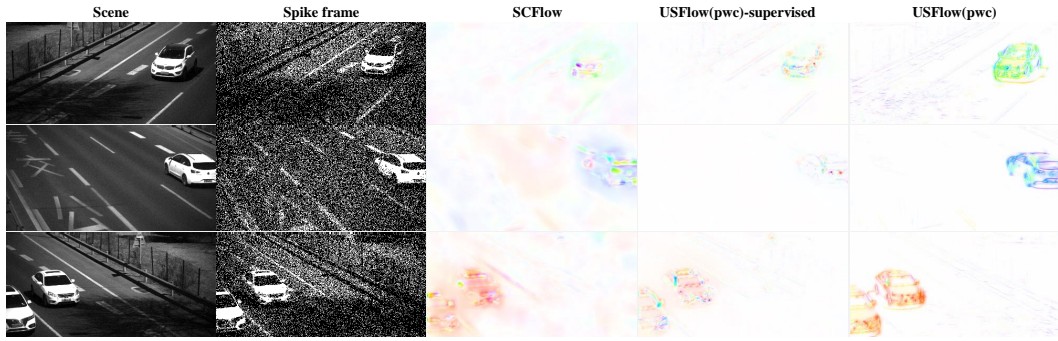

Figure 4: Qualitative results on real scenes. USFlow(pwc) is fine-tuned in the training set of street scenes.

In order to verify that the performance boost is not brought by the number of parameters increasing, we change the dilated convolution to normal convolution with the same input size and feature channel, denoted as RAFT+conv or PWC-Net(variant)+conv. We found that blindly increasing the number of parameters does not make any sense. Normal convolution only provides a limited performance improvement. Therefore, we claim that dilated convolutions can extract salient information more effectively from spike streams. Table 2 shows a comparison between PWC-Net(variant) and USFlow(pwc), indicating that our representation also shows superiority with our unsupervised loss.

**Evaluation of Unsupervised Loss.**  Note that we only build USFlow on PWC-Net(variant) for unsupervised evaluation due to the similar performance of two backbones in Table 1. Since the metric for measuring appearance similarity is critical for any unsupervised optical flow technique [18] in the frame-based domain, we compare with some metrics, *i.e.* the structural similarity index (SSIM) and the Census loss. As illustrated in Table 2, our proposed unsupervised loss can help the model achieve significant performance improvements. Components of our loss are analyzed in Section 5.3.

**Qualitative Results.**  Parts of RGB images, ground truth flow, and the corresponding predicted flow images of the PHM dataset are visualized in Figure 3. Note that PWC(variant) and USFlow(pwc) are trained unsupervised. USFlow(pwc) can predict more detailed flow than PWC-Net(variant) in many regions. However, there still exists a performance gap between the unsupervised method and the supervised method. Moreover, as for supervised methods, the directions of predicted flows (viewed in color) of USFlow(pwc) are closer to the ground truth than SCFlow, especially in object edges.

**Fine-tuned Results.**  We collect some real data in street scenes and split them into training and evaluation set. Due to the advantage of unsupervised learning over supervised learning, we can fine-tune the unsupervised model on the real data, which has no ground truth, to bridge the domain gap. The fine-tuned model has achieved better qualitative results than the supervised model trained on SPIFT in the evaluation set of street scenes. Parts of qualitative results can be found in Figure 4. More clarifications are placed in the appendix.

## 5.3 Ablation Study

**Dynamic Timing Representation.** We do ablation studies on dynamic timing representation in both supervised and unsupervised manners. Table 3 again verifies that dilated convolutions can effectively extract salient information to build promising input representation. More analysis regarding dilated convolutions are in the appendix. Though it seems that layer attention can only improve the performance marginally, we can find out this part makes the training process more stable and faster as illustrated in Figure 6.

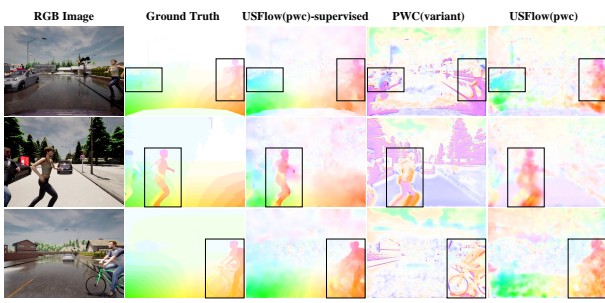

Figure 5: Qualitative results on the SSES dataset. The positions of vehicles and pedestrians are highlighted by black boxes.

**Unsupervised Loss.** Since estimating the light intensity from spike streams is the core of our unsupervised loss, we do ablation studies on it. As shown in Table 4, the mean AEE of the experiment (A) is higher than that of (B). The reason is there are fewer high-speed motion regions than low-speed motion regions in the PHM dataset. Compared with experiments (A) and (B), our unsupervised loss can handle regions with different motion speeds and get the best performance.

## 5.4 SSES Dataset

Based on the CARLA [9], we build a synthetic dataset for extreme scenarios of autonomous driving. CARLA is an open-source simulator for autonomous driving research, which provides open digital assets (urban layouts, buildings, vehicles) to build specific scenarios. Moreover, the simulation platform supports flexible specifications of sensor suites, environmental conditions, and much more. In addition, CARLA can provide the ground truth of optical flow, instance segmentation, and depth.

In the proposed dataset SSES, we design ten extreme scenarios, mainly focusing on traffic accidents caused by violating traffic rules or vision-blind areas. We also include various street scenes, background vehicles, and weather conditions to make scenes more diverse. The demonstration of sample cases and more descriptions of the extreme scenarios are in the appendix.

In all scenarios, the speed setting range is $10 \sim 16$ m/s for cars, $5 \sim 8$ m/s for pedestrians and bicycles, and the frame rates for rendered RGB frames and spike frames are 500 fps and 40K fps respectively. Regarding the generation of spike frames, we first increase the frame rate of RGB frames to 40K fps through a flow-based interpolation method and then generate spikes by treating pixel value as light intensity and simulating the integrate-and-fire mechanism [40]. Note that the ground truth of optical flow is obtained from time aligned with RGB frames. The sequence duration is about $0.5 \sim 1.5s$.

Parts of RGB images, ground truth flow, and the corresponding predicted flow images of the SSES dataset are visualized in Figure 5. USFlow(pwc) can successfully predict the optical flow in regions where vehicles and pedestrians exist (highlighted by black boxes), which can help decision-making in autonomous driving. Table 5 shows the quantitative evaluation of the SSES dataset.

## 6 Conclusions

We propose an unsupervised method for learning optical flow from continuous spike streams. Specifically, we design a dynamic timing representation for spike streams. We also propose an unsupervised loss function in a spike-based manner. Moreover, we simulate extreme scenarios in autonomous driving and propose a validation dataset SSES for testing the robustness of optical flow estimation in high-speed scenes. Experiment results show that our USFlow achieves the state-of-the-art performance on PHM, SSES, and real data.

Table 3: Ablation studies of our design choices for input representation. The present value is the mean AEE, averaged over nine scenes, on PHM. The best results are marked in bold.

| | TMR | LA | Supervised | | Unsupervised | |
|---|---|---|---|---|---|---|
| | | | $\Delta t = 10$ | $\Delta t = 20$ | $\Delta t = 10$ | $\Delta t = 20$ |
| USFlow (pwc) | ✗ | ✗ | 0.943 | 1.797 | 1.783 | 3.336 |
| | ✓ | ✗ | 0.888 | **1.699** | 1.355 | 2.461 |
| | ✓ | ✓ | **0.854** | 1.723 | **1.307** | **2.413** |
| USFlow (raft) | ✗ | ✗ | 0.965 | 1.902 | – | – |
| | ✓ | ✗ | 0.952 | 1.659 | – | – |
| | ✓ | ✓ | **0.931** | **1.649** | – | – |

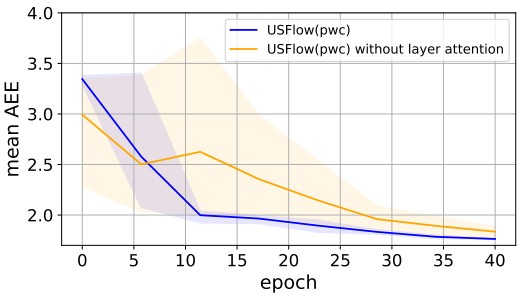

Figure 6: The mean AEE on PHM dataset during supervised training. The shadowed area is enclosed by the min and max values of three training runs, and the solid line in the middle is the mean value.

Table 4: Ablation studies of unsupervised loss. The present value is the mean AEE over the PHM dataset. Best in bold.

| | Setting of Experiment | $\Delta t = 10$ | $\Delta t = 20$ |
|---|---|---|---|
| USFlow (pwc) | (A) Remove $\tilde{I}_T$ from our loss | 2.071 | 3.260 |
| | (B) Remove $\tilde{I}_I$ from our loss | 1.484 | 2.563 |
| | (C) our unsupervised loss | **1.307** | **2.413** |

Table 5: Mean AEE averaged over ten scenes on SSES dataset. All models are trained on the SPIFT dataset.

| | | USFlow(pwc) supervised | USFlow(pwc) unsupervised |
|---|---|---|---|
| mean AEE | $\Delta t = 10$ | 2.967 | 3.122 |
| | $\Delta t = 20$ | 2.234 | 3.130 |

**Limitations.** The characteristics of spike streams generated in extremely dark scenes are quite different from those in bright scenes, so the length of the time window in the unsupervised loss may need to be reset during fine-tuning. We plan to extend our method to address this issue in future work.

# 7  Acknowledgments

This work was supported in part by the National Natural Science Foundation of China under Grants 22127807, 62072009, and in part by the National Key R&D Program of China under Grant 2021YFF0900501.

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
