# OpenReview forum: "Unsupervised Optical Flow Estimation with Dynamic Timing Representation for Spike Camera"
_NeurIPS.cc/2023/Conference — NeurIPS 2023 poster_

### Official Review · Reviewer_PCU5 · 2023-06-10

**Soundness:** 3 good
**Presentation:** 3 good
**Contribution:** 3 good
**Rating:** 6
**Confidence:** 5

**Summary:**

The paper titled "Unsupervised Optical Flow Estimation with Dynamic Timing Representation for Spike Camera" presents a novel approach for unsupervised optical flow estimation using spike cameras. The authors propose a method that leverages dynamic timing representation to estimate optical flow without the need for explicit supervision. They demonstrate the effectiveness of their approach through extensive experiments and comparisons with existing methods.

**Strengths:**

1. The paper introduces a unique and innovative approach to unsupervised optical flow estimation. By utilizing spike cameras and dynamic timing representation, the authors provide a fresh perspective on tackling this problem. This can potentially open new avenues for research and development in the field.
2. The paper presents the methodology in a clear and concise manner. The authors provide sufficient details and explanations, making it easier for readers to understand the technical aspects of their approach. They also include relevant diagrams and figures to aid in comprehension.

**Weaknesses:**

1. The paper lacks a thorough discussion of the limitations of the proposed method. While the authors demonstrate its effectiveness through experiments, it would be beneficial to address potential challenges or scenarios where the method might not perform optimally. This would provide a more comprehensive understanding of the approach.
2. The authors primarily focus on benchmark datasets and quantitative evaluations, but they do not provide real-world examples or scenarios where their method could be applied. Including such examples would enhance the practical relevance of the research and highlight its potential impact.

**Questions:**

1.How does the proposed method handle occlusions or complex scenes where multiple objects are moving simultaneously? Are there any specific limitations or challenges in such scenarios?
2. How does the dynamic timing representation used in the spike camera contribute to the accuracy of optical flow estimation compared to traditional frame-based approaches?


**Limitations:**

The proposed method's performance and generalizability might be limited to specific datasets or conditions. It would be valuable to explore its effectiveness on a wider range of datasets, including more challenging and diverse scenarios.

---

> ### Author Rebuttal · Authors · 2023-08-10
>
> Thank you for your precious time and reviews.
>
> **Q1. Lack a thorough discussion of the limitations.**\
> We fix the data length of spike streams which participate in the calculation of our unsupervised loss function. In scenes with normal brightness, this fixed-length spike data can provide sufficient information for inferring light intensity. However, the intervals between spikes are relatively large in the ultra-dark scenes, and light intensity cannot be accurately inferred using the spike streams of this fixed data length. We note relevant examples in real-world data. For example, when estimating the optical flow for a dark car in the shadow of an overpass, we note that different positions of the car have different optical flows. In future work, we plan to perform interval analysis on the input spike data and dynamically select the appropriate-length spike data to participate in the unsupervised loss function according to the statistical distribution of the interval length.
>
> **Q2. The author does not provide real-world examples or scenarios where their method could be applied.**\
> Our method is suitable for high-speed motion scenes with normal brightness in the real world. The Figure 4 shows the test results of USFlow and comparison methods on the real spike data captured by a spike camera in real-world. The shooting device we use is the first-generation spike camera with a spatial resolution of 250 × 400 and a frequency of 20 kHz. For these real spike data, there are more detailed descriptions in the supplementary material. However, in outdoor scenes, obtaining the ground truth of optical flow can only rely on LiDAR, and the ground truth collected by this way has measurement and calibration errors. We cannot collect the accurate ground truth in these real street scenes. We can only demonstrate the performance of our method by visually comparing it with other spike-based methods on these real spike data.
>
> **Q3. How to handle the occlusions or complex scenes where multiple objects are moving simultaneously? Are there any specific limitations or challenges?**\
> The spike camera has an ultra-high frequency, and it can continuously record high-speed motion scenes. In the application of the spike camera, complex motion is usually divided into multiple small motions along the time dimension. Note that the time span corresponding to each small motion is very small, so the occluded area changes very little in each small time span. Then the spike data within these short time spans are used to estimate the corresponding small motions respectively. Finally, all these estimated small motions make up the complete complex motion. Therefore, the influence of occlusion problem in spike-based methods is less than in frame-based methods, and we did not design a particular structure for this problem. The proposed method currently does not deal well with the problem of occlusion caused by the simultaneous movement of multiple objects. It addresses the main issues of the spike-based methods: the effective representation of spike data and the lack of sufficient datasets with ground truth.  In future works, we will consider designing a structure to handle the occlusion issue.
>
> **Q4. Compared to frame-based methods, how does the dynamic timing representation used in the spike camera contribute to the accuracy of optical flow estimation?**\
> Compared with traditional cameras, spike camera has an ultra-high frequency and the ability to continuously record high-speed motion with a high temporal resolution. Given two adjacent RGB frames, frame-based methods usually model the motion between two RGB frames as linear motion. However, with the help of the continuous spike data between these two frames, the motion to be estimated can be divided into multiple small motions along the time dimension to estimate separately. These estimated small motions are then used to compose the complete motion. Therefore, nonlinear and non-uniform complex motions can be estimated by using spike data. For spike-based methods, how to represent the spike data is the key point because the data structure of spike data is different from the data structure of RGB frames. In spike representation, selecting an appropriate data length is pivotal. A too-long spike stream is not suitable for high-speed regions since time offset accumulation introduces more redundant information and a too-short cannot exhibit light intensity precisely with few binary data. The proposed Dynamic timing representation breaks the limitation of the fixed time window and it dynamically selects the spike data of the appropriate length according to the motion speed at each pixel position for representation. It improves the quality of representation and thus the accuracy of optical flow.
>
> **Q5. It would be valuable to explore its effectiveness on a wider range of datasets.**\
> The benchmark dataset used in the experiments for testing is the PHM dataset which is generated by a graphics-based simulator. It is the most common dataset in the spike-based optical flow estimation task. In other tasks, there are some spike datasets, in which the spike data is generated by simulating the working mechanism of the spike camera based on RGB frame sequences. The frame rates of these RGB frame sequences are much lower than that of spike camera, so the motion information between two adjacent RGB frames can only be generated by frame-based methods. Thus, there is usually a difference between the real spike data captured by the spike camera and the simulated spike data generated by algorithms simulating the spike camera working mechanism. In order to verify the effectiveness of the proposed method on a wider range of data, we use a spike camera to capture a series of real spike data from different angles in the real world. Then we test and compare the performances of methods on these real spike data. The visualization results are shown in Figure 4 of our paper.

---

> > ### Comment · Reviewer_PCU5 · 2023-08-10
> > **Thanks for the responses**
> >
> > I really appreciate the responses. The authors have successfully addressed all of my concerns. While I am happy to raise the score, I have to stress that the paper focuses more on an engineering problem rather than a generic algorithmic challenge. I think this manuscript can be a very strong ICRA/IROS paper but may not provide much value in AI community. With that said, my conclusion is weak acceptance.

---

> > > ### Author Response · Authors · 2023-08-11
> > > **Thank you for your response**
> > >
> > > We are glad to address all your concerns successfully. Thank you for your insightful comments and raising your score to borderline acceptance.

---

> > > > ### Comment · Reviewer_PCU5 · 2023-08-11
> > > >
> > > > Also, I think it would be very interesting to theoretically compare it to a state-of-the-art method, such as 'Transflow: Transformer as a Flow Learner [CVPR 2023]'. This may provide insight for the proposed method in relation to concurrent work.

---

> > > > > ### Author Response · Authors · 2023-08-11
> > > > > **Thank your for your advice**
> > > > >
> > > > > TransFlow is a related and interesting work. We will discuss TransFlow in related works. The idea of utilizing the long-range temporal association to recover more information ($\textit{e.g.}$, occlusion) in TransFlow provides valuable insights into addressing the occlusion issue within spike-based optical flow estimation. However, there are some differences between TransFlow and USFlow. The TransFlow focuses on improving the accuracy of correlations to estimate high-quality optical flow. While based on the characteristics of spike data, USFlow focuses on effectively and accurately representing spike data and designing a new unsupervised loss in spike-based optical flow estimation. From the perspective of data flow, there are differences in the strategies for constructing and utilizing correlations between TransFlow and USFlow. The TransFlow additionally uses the long-range temporal associations and utilizes Multi-head Self-Attention (MSA) and Multi-head Cross-Attention (MCA) to construct the correlations. Then it estimates the optical flow from the correlations. USFlow firstly represents spike data by the proposed Dynamic Timing Representation (DTR) module. Then it uses CNNs to construct correlations from these spike representations. Eventually, USFlow gradually refines the estimated optical flow under the guidance of correlations.

---

> > > > > > ### Comment · Reviewer_PCU5 · 2023-08-18
> > > > > >
> > > > > > Looking forward to it. Good luck!

---

### Official Review · Reviewer_Hpzy · 2023-06-28

**Soundness:** 2 fair
**Presentation:** 1 poor
**Contribution:** 2 fair
**Rating:** 6
**Confidence:** 4

**Summary:**

The paper proposes a method for estimating optical flow from event streams captured with a spike camera. The approach is based on unsupervised learning and achieves good results in the comparison to other works in the SotA, including conventional and event-based approaches. Authors collect their own dataset and evaluate on it, specially, in an attempt to show the advantage of finely estimating for low and high-speed regions (although, this is not clearly demonstrated with the current experimental section). Finally, authors also include a section with an ablation study to assess the impact of the different parts of the model (loss terms) and the supervised vs. unsupervised learning approaches.

**Strengths:**

+ Authors prepare and collect a dataset with is always a hard but valuable task
+ The unsupervised learning approach provides a valuable alternative for optical flow estimation and shows good results on the evaluated datasets
+ It is unsupervised, which is another good keypoint to bear in mind
+ The ablation study on the loss terms impact and the comparison between supervised vs. unsupervised are very valuable

**Weaknesses:**

- After reading the work, I don't think authors make clear the differences between spike and event cameras. There is a confusion all over the introduction and methodology and this makes me (and the reader) hesitate about the relevance of this work.
- The work needs extensive English proofreading, some sentences are hard to understand.
- The conclusions should not be a summary of the work
- In the experimental section, authors compared on PHM and propose their own dataset but they are not comparing to the widely used MVSEC and DSEC datasets (that are the most common for event-based processing). This will improve not only completeness but also the comparison to other works in the state of the art.
- From the paper and since authors propose an artificially generated dataset, it is not clear if 1) the spike camera is a real sensor and 2) in case it is, if they have access to real sequences recorded with it.


**Questions:**

- Regarding the difference between event and spike cameras. Authors should make clear what kind of data they are handling. Event cameras produce asynchronous events and in fact, one of the advantages of these cameras is the lack of need for reconstructing the dynamics of the scene due to their high temporal resolution. This is great e.g. for optical flow estimation. However, it seems that spike cameras produce frames at very high resolution (something closer to a high-speed camera). This could also be interesting but it carries the burden of analyzing a lot of full frames, which cannot be ignored as a drawback. I really think that, since they are comparing to both conventional and event cameras, this should be clarified as soon as possible in the paper.
In fact, this is mixed up in the paper, for example, at some point authors mention something about spike frames but then, they point out that pixels respond asynchronously (?)
- When comparing to event camera processing, performance should be part of the comparison as well (or latency).
- In the abstract, lns 12-14 do not make reference to the dataset (I think this should be added here as well)
- In page 2, I believe the correct term is luminance (not illuminance)
- In line 68, authors mention "shows visually impressive performance", what do you mean? Is that qualitative results? I guess authors mean that they are appealing? Are they only qualitative?
- The sentence in Lns 107-108 seems incomplete
- In Ln 182, are 40 and 100 ms? or frames?
- In ln. 196 "different numbers of intervals"
- Lsmooth is not defined
- There are some recent works that use SNNs such as "Optical flow estimation from event-based cameras and spiking neural networks", from J. Cuadrado et al. that could help to improve the state of the art section.
- Citations are weirdly formatted. Authors should make sure that they are using the conference format
- In section 5.3 authors (unsupervised loss) authors mention Figure 4 when they mean Table 4.
- What is the contribution of the LA module? According to Fig. 6 it seems it is not helping at all ... in such a case, why do authors keep it? is there another case in which the LA module does add a significant contribution?
- At the end of section 5.4, authors discuss results on SSES dataset but they mention before that they are also doing some real-world sequences, where are the results on these sequences?

**Limitations:**

The work includes a limitations section that seems adequate

---

> ### Author Rebuttal · Authors · 2023-08-10
>
> Thank you for your precious time and reviews.
>
> **Q1. Authors do not make clear the differences between spike and event camera.**\
> Both spike camera and event camera have ultra-high frequency, and they can continuously record high-speed motions. However, the working mechanism of them are different. In the spike camera, when the accumulated photons reach a preset threshold, a spike will be fired. The event camera records the light intensity change information on the corresponding pixels whose light intensity changes exceed the threshold. Spike camera and event camera have their own application advantages. In the analysis of high-speed motion scenes, there are indeed some related event-based methods. The method proposed in this paper uses spike data output by spike camera to estimate optical flow and it belongs to the spike-based method.
>
> **Q2. Authors should make clear what kind of data they are handling. Spike camera carries the burden of analyzing a lot of full frames.**\
> The proposed method uses the spike data output by the spike camera to estimate optical flow. Spike camera and event camera have their own advantages. The event camera records motion information according to the relative changes in light intensity and outputs relatively sparse event streams. Thus, event camera has great application value in tasks such as motion estimation, and objects tracking. The spike camera outputs binary spike data at each pixel position according to photon accumulation. This full-pixel recording feature makes spike camera suitable for pixel-dense tasks, for example, high-speed motion scene reconstruction. When using a spike camera to reconstruct a high-speed motion scene, optical flow estimation is usually involved, so there is a need to estimate optical flow based on spike data. Thank you for your suggestions. We will clarify in the paper that spike camera records scene information at full resolution.
>
> **Q3. In the experimental section, authors compared on PHM and propose their own dataset but they are not comparing to the widely used MVSEC and DSEC datasets (that are the most common for event-based processing).**\
> This paper focuses on using spike data which is output by the spike camera to estimate optical flow. The input of the whole model is spike data. The MVSEC and DSEC datasets do not contain spike data, so these two datasets cannot be used as test datasets for the spike-based methods.
>
> **Q4. if the spike camera is a real sensor? In case it is, if they have access to real sequences recorded with it.**\
> The spike camera is a real camera. We use the first-generation spike camera (20 kHz, 250 pixels × 400 pixels) to collect a series of real spike data in real-world. In our paper, Figure 4 shows some test results on these real spike data. The SSES dataset is generated by CARLA simulator. We plan to release all new datasets used in our paper in the future.
>
> **Q5. When comparing to event camera processing, performance should be part of the comparison as well (or latency).**\
> It is not the goal of this paper to compare spike camera with event camera in optical flow estimation. Since spike camera applications usually involve motion estimation, this paper aims to study how to estimate more accurate optical flow from spike data which is output by spike camera.
>
> **Q6. In the abstract, lns 12-14 do not make reference to the dataset.**\
> We will make reference to PHM dataset here.
>
> **Q7. What does "shows visually impressive performance" mean?**\
> It means the visualization results of USFlow on real spike data are significantly better than other methods.
>
> **Q8. The sentence in Lns 107-108 seems incomplete.**\
> At the end of the subsection that introduces various types of representation works, Lns 107-108 is a preview of the proposed representation method.
>
> **Q9. In Ln 182, are 40 and 100 ms? or frames?**\
> 40 and 100 represent 40 spike frames and 100 spike frames, respectively.
>
> **Q10. What does "different numbers of intervals" mean?**\
> The interval is defined in Lns 186-188. Multiple intervals are spliced to form a time span, and we use multiple such time spans to infer light intensity. These time spans contain (2k-1) intervals, the k is a hyper-parameters in experiments.
>
> **Q11. Smooth loss function is not defined.**\
> The $L{smooth}$ used in our method is a basic version which is defined as: $\mathcal{L}{\rm smooth}\left( f, f' \right) = \frac{1}{HW} \sum_{\mathbf{x}} \vert \nabla f (\mathbf{x}) \vert + \vert \nabla f' (\mathbf{x}) \vert$, where $f$ and $f'$ are bidirectional estimated optical flow, $( \nabla )$ is the difference operator, H is the height and W is the width of the optical flow.
>
> **Q12. There are some recent works that use SNNs. These works could help to improve the state-of-the-art section.**\
> The proposed USFlow estimates optical flow based on spike data. Although the output of spike camera and event camera have different data structures, some event-based methods and SNN works may indeed be instructive. We will keep our attention on such methods.
>
> **Q13. What is the contribution of LA module?**\
> Although the LA module cannot bring a huge improvement in performance, it can make the network converge faster and the training process more stable.
>
> **Q14. Where are the results on real spike data captured by the spike camera?**\
> In Figure 4, we show some results on real spike data. We capture these real spike data by a first-generation spike camera (20 kHz, 250 pixels × 400 pixels) from different angles.
>
> **Q15. Writing advice.**\
> Thanks for your concern. We will correct these issues.

---

> > ### Comment · Reviewer_Hpzy · 2023-08-12
> > **Thanks for the rebuttal**
> >
> > I thank the authors for their rebuttal. Now, I have a clearer picture of the work. I think authors should carefully proofread the work for the final version. Also, and regarding the mentioned SNN-based works, I suggest some recent ones such as:
> > - Optical flow estimation from event-based cameras and spiking neural networks
> > - Taming Contrast Maximization for Learning Sequential, Low-latency, Event-based Optical Flow

---

> > > ### Author Response · Authors · 2023-08-12
> > > **Thank you for your response**
> > >
> > > Thank you for your valuable suggestion. We will certainly add discussions of these works in our paper.

---

### Official Review · Reviewer_2gct · 2023-06-29

**Soundness:** 2 fair
**Presentation:** 3 good
**Contribution:** 3 good
**Rating:** 4
**Confidence:** 5

**Summary:**

This paper propose an unsupervised learning framework for spike-based optical flow estimation, which is mainly developed for spike input representation and spike loss function. In general, I think the method has merit, but the experiment is unconvincing in its lack of clarity.

**Strengths:**

1. Propose a lightweight end-to-end spike data representation with the function of a dynamic time window.
2. Propose a two-stage unsupervised loss to model light intensity in regions with different motion speeds.

**Weaknesses:**

1. In Fig. 5, there are significant quantization/discontinuities in the optical flow gt of the SSES dataset, concentrated in the ground area. To my knowledge, this is not present in other optical flow datasets. The reliability of the optical flow ground-truth is a concern, please provide an explanation.
2. In Fig. 4 and supplementary video, the results of these methods using only spike as input look bad. The major problem is that the results are concentrated only the textures and are not consistent for the same object., which cannot be called ``dense`` optical flow. To my knowledge, this phenomenon is not present in unsupervised two-frame based optical flow  estimation methods. I suggest comparing the outputs of two-frame based methods and analyzing the spike-based failure cases.
3. The performance of using spike cameras for optical flow estimation is of concern. In L27, spike camera can record details of high-speed moving objects, it has enormous potential for estimating more accurate optical flow in high-speed scenes. But comparing the state-of-the-art optical flow estimation methods using image cameras, I believe this claim is not convincing. The visualization results of existing spike-based methods, including the authors', are significantly worse than existing image-based methods.

**Questions:**

1. L158, Blindly fusing the output of all layers may impair the learning process. It is desirable to have experiments to support this belief.
4. L170 seems duplicate to L122.
5. I suggest to complete the experiments of raft backbone model.

**Limitations:**

1. L132, advantage of neuromorphic vision is low latency. However, the authors do not present the runtime analysis of the proposed method.
2. See weaknesses above.

---

> ### Author Rebuttal · Authors · 2023-08-10
>
> Thank you for your precious time and reviews.
>
> **Q1. About the reliability of the optical flow ground-truth.**\
> The SSES dataset is a verification dataset containing various corner cases in the autonomous driving field and it is generated by CARLA. CARLA is an open-source simulator for autonomous driving research and it can provide the ground truth of optical flow. The proposed SSES dataset is used to verify the effectiveness of the methods in extreme cases. However, the temporal resolution of spike data is ultra-high, which results in relatively small motions between spike frames. Therefore, although the generated ground truth is a smooth optical flow field, it is limited by the quantization accuracy of the CARLA, and quantization appears in the visualization of the ground truth. For high-speed motion scenes, collecting optical flow by using LiDAR can circumvent the need for using simulators. However, the frequency of LiDAR is relatively low, and the process of calibration and measurement can introduce larger errors to the ground truth.
>
> **Q2. The performance of spike-based optical flow estimation methods is not better than frame-based methods.**\
> The frequency of spike cameras can reach up to 40 kHz, so spike cameras can continuously record high-speed motion scenes. The spike camera serves to compensate for the drawbacks of traditional cameras in high-speed motion scenes, where information loss arises from insufficient frame rates. If there is complex nonlinear and non-uniform motion between two adjacent RGB frames, the common frame-based methods usually approximate the complex motion as linear motion. However, with the help of spike data containing continuous motion information, the complex motion can be estimated. Specifically, the spike data containing the information of complex motion is divided into multiple spike data with shorter data lengths. Then corresponding small motions can be estimated from these spike data with shorter data lengths. Finally, all these small motions are merged into a complete complex nonlinear and non-uniform motion.
>
> **Q3. Questions about the quality of the results shown in Figure4.**\
> Directly estimating optical flow on spike data using unsupervised learning is a new and difficult topic. It is very challenging to infer light intensity from spike data, because it is necessary to consider many factors, such as the brightness of scenes and the motion speed of different objects. Thence, directly estimating optical flow on spike data is more challenging than estimating optical flow on RGB images. The data corresponding to the Figure 4 is captured by a spike camera in the real-world and it only contains spike data. Limited by the frame rate of traditional camera, we don’t have RGB frames corresponding to these real spike data. The scenes on the left column of Figure 4 are reconstructed by a professional reconstruction method, and these reconstructed RGB images contain more noise than real RGB images. When these reconstructed RGB images are used in frame-based methods, the accuracy of estimation will be affected. Thence, the real spike data corresponding to Figure 4 cannot be used to compare the performance of spike-based methods and frame-based methods. We just reconstruct the scenes to tell the readers what contents are in the scenes. In the practical applications of spike cameras, there is only spike data can be used.  As shown in Figure 4, the proposed method shows significant advantages over other spike-based methods, and it also promotes the exploration of unsupervised learning in spike-based optical flow estimation task. For spike-based unsupervised methods, there is still space for exploration. We may further study this issue in our future works.
>
> **Q4. It is desirable to have experiments to support L58.**\
> The "blindly fusing" here means not using layer attention, but simply averaging the output of all layers. In the ablation study on the layer attention module, simply averaging all layer outputs is used to replace the layer attention module. As shown by the orange curve in **Figure 6**, it can be seen that simply averaging all layer outputs does impair the learning process.
>
> **Q5. L170 seems duplicate to L122.**\
> In order to start the discussion of unsupervised loss function design, L170 briefly recalls the problem setting and clarifies again that the proposed method doesn’t need ground truth in training. Thank you for your advice, we will simplify the expression.
>
> **Q6. Suggest to complete the experiments of raft backbone model.**\
> We test the USFlow(raft) which is trained by  unsupervised learning on PHM dataset, and AEEs are shown below. As indicated by Table 2, it can be observed that both USFlow(raft) and USFlow(pwc) show the similar performance.
> |  $\Delta$ t |  method  |  Ball  |  Cook  |  Dice  |  Doll  |  Fan  |  Hand  |  Jump  |  Poker  |  Top  |  Mean  |
> | :--------: | :--------: | :--------: | :--------: | :--------: | :--------: | :--------: | :--------: | :--------: | :--------: | :--------: | :--------: |
> | $\Delta$ t=10 | USFlow(raft) | 0.760 | 1.902 | 1.923 | 0.631 | 0.724 | 2.439 | 0.490 | 1.217 | 2.556 | 1.405 |
> | $\Delta$ t=20 | USFlow(raft) | 1.249 | 3.507 | 2.980 | 0.964 | 1.112 | 3.440 | 0.624 | 2.216 | 4.628 | 2.303 |
>
> **Q7. About runtime.**\
> During the process of designing the network, one of our goals is to shorten the runtime by designing a lightweight model. Estimating the optical flow between two timestamps on a 3090 Ti GPU, the runtime of our proposed USFlow is 90.6ms, and the runtime of the comparison method SCFlow is 236.9ms.

---

> > ### Comment · Reviewer_2gct · 2023-08-12
> >
> > I have read the authors' responses as well as the comments of other reviewers. I agree that the methods are somewhat contributing, but I still maintain that the evaluation presented in this paper is limited, so I will lower my rating.
> > I believe that all three of the weaknesses that concerned me have not been definitively addressed by the authors.
> > For Q1, is the simulated optical flow reliable? The authors mentioned the limited quantization accuracy of CARLA as a problem, so does this problem still exist with the optical flow data stored in the simulated dataset? I think a conclusive answer is needed.
> > For Q2, the authors start the paper by claiming that the spike camera has enormous potential for estimating more accurate optical flow in high-speed scenes (from line 26). But the results presented in their paper fall short of my expectations for the performance of state-of-the-art optical flow estimation methods. The authors do not answer why the optical flow estimated from spike data is not spatially dense.
> > For Q3, the reason given by the authors for not comparing the two-frame method is the noise contained in the images, which does not convince me. From Figure 4, I think the image quality is acceptable and I believe that the existing two-frame methods can easily provide better optical flow estimation results.
> >
> > In addition, I think 90ms is still slow for low latency neuromorphic vision, but of course this is not a problem that should be addressed by every frontier study.

---

> > > ### Author Response · Authors · 2023-08-12
> > > **Thank you for your response**
> > >
> > > We will give clearer clarifications to your concerns below.
> > >
> > > For the concern of Q1, we checked the unique values of the gt, sorted them, and looked at the diff between each unique value. The diff is exactly 1/1024 which points to a 10-bit quantization. Thence, the dataset is reliable, and its accuracy is 1/1024. We will release this dataset for others to check.
> > >
> > > For the concern of Q2 and Q3, we would change the claim to $\textit{it has enormous potential for estimating optical flow in high-speed scenes}$. We admit some existing two-frame methods can provide better optical flow estimation results.
> > >
> > > However, we want to clarify that when the motion to be estimated occurs during an extremely short time, the conventional RGB camera cannot provide the corresponding RBG frames due to its low frequency.   In other words, If we cannot obtain data in some high-speed scenarios, it is hard to obtain optical flow. Hence, the spike camera and conventional RGB cameras have different application scenarios.
> > >
> > > Spike camera is only a complement for the various vision tasks. Since it is not a replacement for RBG camera, it cannot surpass the frame-based work in all scenarios.
> > >
> > > In addition, as a counterpart, event camera also cannot surpass the RBG camera in optical flow estimation if cameras both have data for a certain scenario. However, it doesn’t mean there is no meaning to investigating this direction and it is indeed a thriving field at the current stage.
> > >
> > > The unsupervised finetune dataset is small, which leads to the phenomenon that the estimated optical flow is not spatially dense. Note that the results are spatially dense in Figure 3.
> > >
> > > As mentioned by other reviewer, unsupervised spike-based optical flow estimation is a new venue. However, the unsupervised frame-based methods have been studied for a long age, and they have many better training datasets which can boost the generalization. We admit at the current stage, our method has limitations, but our results look not bad and are supported by other reviewers. In addition, only when this new venue is encouraged, more robust datasets and more works can emerge to promote the development of this field.

---

> > > ### Comment · Reviewer_PCU5 · 2023-08-12
> > > **Why lower your rating?**
> > >
> > > Reviewer 2gct,
> > > Thank you for your diligent review efforts. I am currently engaged in a thorough examination of your comments and find myself engaged in a process of comprehending the underlying rationale behind your suggestion to "lower my rating." Should the authors have encountered challenges in adequately addressing your inquiries, it is reasonable to consider our initial evaluation as the benchmark. Conversely, any diminution of the rating is justifiable only in cases where the authors have, inappropriately, provided responses that are contrary to the essence of your inquiries.
> > >
> > > Upon careful scrutiny, I am inclined to assert that the authors have, to the best of my observation, diligently addressed the issues raised in your comments, offering responses that are in alignment with the nature of your inquiries. Consequently, it is my contention that the grounds for reducing the rating appear to lack a sufficiently robust foundation.
> > >
> > > It is essential to underscore the significance of fostering a constructive discourse within the academic community to facilitate the cultivation of insightful new knowledge. As many of us are actively engaged in various capacities, encompassing roles as authors, reviewers, and ACs, we bear a collective responsibility to uphold the principles of impartial and judicious assessment that are integral to the progress of our academic community.
> > >
> > > This communication serves as a respectful reminder of our shared commitment to the responsibilities inherent in the role of a reviewer. Its intent is in no manner geared towards meddling with the established rating, but rather serves as a courteous prompt to reiterate our shared role in promoting a culture of conscientious and equitable evaluation.
> > >
> > > Thanks for reading my message.

---

### Official Review · Reviewer_eenq · 2023-07-03

**Soundness:** 3 good
**Presentation:** 3 good
**Contribution:** 3 good
**Rating:** 6
**Confidence:** 3

**Summary:**

This paper works on the optical flow estimation problem, especially in the unsupervised setting, for high-frequency spike camera inputs. Specifically, a dynamic timing representation module and a spike-based unsupervised loss are proposed to improve performance. A new synthetic dataset, namely SSES, is created to test extreme scenes in autonomous driving. Experiment results show the proposed method has achieved state-of-the-art performance compared with previous methods.


**Strengths:**

1. The paper is well-motivated by introducing the importance of optical flow estimation for high-frequency inputs collected from latest developed sensors like spike cameras. This topic has not been extensively studied so far.
2. The paper is overall clear and well-structured.
3. The proposed method is reasonable. Explanations on each module design are provided.
4. The results are compelling.


**Weaknesses:**

1. There are still some confusions in the text or figures that need to be clarified. See additional comments below.
2. It will be better to polish the language of the paper before publication. Correct grammatical errors and use more formal language. Avoid informal expressions like "a bunch of" (Line 138), "won't" (Line 140), etc. See additional comments below.
3. Authors are encouraged to share their code and generated datasets for better reproducibility. Such plan has not been declared in the paper.


**Questions:**

1. If I understand it correctly, the method uses the predicted optical flow to determine whether each pixel belongs to a low-speed or high-speed region and then uses different ways to approximate light intensities $\tilde{I}$. At the starting iterations of training, the predictions could be totally off, so it is likely that the method will pick a wrong way to approximate intensities. Did you try to tackle this issue?
2. The data used in Fig 4 do not have ground-truth labels, right? Only qualitative evaluation is possible, so this evaluation may be weak.
3. There is also a large part of supervised results, so stating "unsupervised" in the title could be misleading.
4. I am curious how a normal RGB-based optical flow network would perform as a comparison. For example, a naïve approach is to first convert spike inputs to RGB and then apply a state-of-the-art RGB-based optical flow network of similar size (parameters, memory cost). What would the results be like? This could be a baseline to strengthen the motivation of developing optical flow networks specific to spike camera inputs.



Additional comments:
1. Grammatical errors: run-on sentences (Line 103-105, line 140-141, Line 163-165, Line 310-311).
2. Eq 1: The "mod" operator is traditionally only defined for integer division, but the value here is clearly continuous numbers, so it is better to define "mod" in the text in the continuous context.
3. Chapter 3.1: There are some confusions here. (1) I assume $A(\mathbf x, t)$ is continuous, but Line 119 says the output stream is binary, so there is missing gap in between that need explanations. Maybe adding a definition, something like $S_t^N(\mathbf x) = \mathbb I(\sup_{\tau\in[t-\Delta t, t+\Delta t]} A(\mathbf x, \tau) =\theta)$ will make it clear. (2) Line 113-114 say the camera "fires a spike" when the cumulative electrons hits the threshold, so I assume it is asynchronous for each pixel, but then Line 117-118 say the camera "fires a spike" at time T, which is synchronous and periodic. Maybe you could change the expression here to avoid confusion.
4. Line 130: "officially" -> "official".
5. Line 135: "dependence" -> "dependency".
6. Fig 2: Text is too small. It is better to make text size close to that of the main paper so that readers can read the figure clearly without needing to zoom in.
7. Line 147: It is better to add that your convolution is 1D on the time dimension to avoid confusion.
8. Line 182-183: Does your method guarantee $\omega_s + \omega_l = 1$? Eq 6: Does $\sum_k\omega_k=1$? If so, state so explicitly.
9. Eq 7 could be merged into the text. It does not have to an equation.
10. Line 175-184: What do you mean by "speed"? The frequency of camera readings, or fast motion? Please clarify.
11. Line 207: The notation $\omega_s, \omega_l, \omega_{k=1}, \omega_{k=2}$ is messy. Try not to use both textual abbreviations ("s", "l") and values ($k$) as subscripts for the same letter $\omega$. Use other letters like $\alpha_s, \alpha_l$, $\beta_k$ instead; or, you could do superscripts with parentheses $\omega^{(s)}, \omega^{(l)}, \omega_k$.
12. Line 209: "approximate" -> "approximated".
13. Line 212: define or cite "Charbonnier loss"; Line 213: define or cite "smoothness loss" specifically. There are many ways to define "smoothness".
14. Table 1 and 2: It is better to add citations into the table, so we know which data are generated from your experiments and which data are cited from other papers. The current table assumes that all data are credited to yourself.
15. Table 1 and 2: I recommend adding a vertical line between the "Top" and "Mean" columns, so we know "Top" is the name of the scene instead of the top error across all scenes.
16. Line 277: "through" -> "though".
17. Line 284: "Figure 4" -> "Table 4".

**Limitations:**

Yes

---

> ### Author Rebuttal · Authors · 2023-08-10
>
> Thank you for your precious time and reviews.
>
> **Q1. Do authors have plan to share their code and generated datasets ?**\
> We will release the code, models and datasets.
>
> **Q2. The problem about the network may choose wrong way to approximate light intensity at the starting iterations of training.**\
> In high-speed regions, the light intensity estimated by the method which is suitable for low-speed regions will be affected by motion blur. In low-speed regions, the light intensity estimated by the method which is suitable for high-speed regions will contain noise. While the quality of these light intensities estimated in the wrong ways is not good, they are still sufficient for initiating the training of the network.
>
> **Q3. The real spike streams output by spike camera lack ground truth.**\
> There are two approaches to obtaining the ground truth of optical flow from real-world data: 1. The first method is used for indoor scenes. The optical flow can be obtained with the assistance of ultraviolet (UV) light. 2. The second method is used for outdoor scenes. It uses LiDAR to collect optical flow. However, this method will introduce calibration and measurement errors, and the collected optical flow field is sparse. Obtaining the ground truth of optical flow in outdoor scenes is challenging. Therefore, we show the performance of USFlow by comparing the visualization results of USFlow and other methods on real spike data.
>
> **Q4. The paper shows some supervised results, so stating "unsupervised" in the title could be misleading.**\
> In Table 1, the comparative methods are all supervised methods. Therefore, in Table 1, we also employ supervised training in order to compare the performance of the representation module. The method presented in Table 1 is not the final method proposed in this paper. After validating the effectiveness of our representation module in Table 1, we proceed to Table 2 where our USFlow is trained by unsupervised learning. Our proposed method includes a novel unsupervised loss function, and the USFlow trained using unsupervised learning is our proposed method in this paper.
>
> **Q5. First convert spike inputs to RGB and then apply SOTA RGB-based optical flow estimation networks. What would the results be like?**\
> Reconstructing spike data into RGB images is currently a developing research topic, involving intricate considerations. Simple principle-based methods for reconstructing RGB images from spike data often have issues with noise and blurriness. Even the state-of-the-art frame-based optical flow estimation methods use such poor-quality reconstructed RGB images as input, the accuracy of the estimated optical flow remains challenging. A well-designed reconstruction method is highly complex, such as Zhao's Spk2imgnet, which needs to consider many complex factors, and the reconstruction process involves a motion analysis module. First reconstructing the spike data into RGB images and then estimating the optical flow is a kind of solution with a large space for exploration. Our proposed method which directly estimates optical flow on spike data belongs to another kind of solution.
>
> **Q6. In L175 and L184, what do you mean by "speed"?**\
> The "speed" at here means motion speed.
>
> **Q7. "Charbonnier loss" and "smoothness loss".**\
> The Charbonnier loss we use is the basic version: $\\rho(x)=(x^2+\epsilon^2)^r\$. The smoothness loss we use is also a simple version which is defined as: $\mathcal{L}{\rm smooth}\left( f, f' \right) = \frac{1}{HW} \sum{\mathbf{x}} \vert \nabla f (\mathbf{x}) \vert + \vert \nabla f' (\mathbf{x}) \vert$, where $f$ and $f'$ are bidirectional estimated optical flow, $( \nabla )$ is the difference operator, H is the height and W is the width of the optical flow. We will add definitions about them.
>
> **Q8. Additional comments.**\
> Thank you for your concern. We will correct grammatical errors, add citations in tables and clarify confusion.

---

> > ### Comment · Reviewer_eenq · 2023-08-10
> >
> > Thank you for your response. I don't have more questions for now.

---

> > > ### Author Response · Authors · 2023-08-11
> > >
> > > Thank you for your approval of our work.

---

### Official Review · Reviewer_fVdD · 2023-07-05

**Soundness:** 2 fair
**Presentation:** 3 good
**Contribution:** 2 fair
**Rating:** 6
**Confidence:** 3

**Summary:**

The paper introduces a method for optical flow estimaiton for spike cameras. Because of the high temporal resolution of the camera, a preprocessing step based on temporal dilated convolution and attention layers is used to automatically select the best temporal scale for a given sequence.
The authors also introduce a unsupervised loss based on the reconstruction of intensity values form spike data combined with the photometric loss.


**Strengths:**

The main strengths of the paper are the temporal dilated convolution preprocessing that allow for automatic temporal scale selection, as well as the unsupervised loss.
The description of the method is clear and the experimental section complete, including ablation studies and comparision with previous works.



**Weaknesses:**

The paper does not address the question of latency and computational time, which are key aspect of the spike camera.
Also, since frames are available in the simulate datasets it would have been interesting to compare the accuracy of the proposed method on standard rgb frames, to quantify the benefits of using a high temporal resolution sensor for optical flow estimation.
Finally, for people not familiar with the spike camera, it would be beneficial a more complete introduction to this type of sensor. There is no reference in the paper to any publications explaining what a spike camera is, it's HW implementation and carachteristics.


**Questions:**

please refer to "weaknesess" section

**Limitations:**

The authors could add a general discussion of advantages and limitations of the spike camera and of optical flow from spike cameras.
For example any the challenge related to data rate, computational time and power consumption of these devices.

---

> ### Author Rebuttal · Authors · 2023-08-10
>
> Thank you for your precious time and reviews.
>
> **Q1.The latency and computational time of the proposed method.**\
> In USFlow, to estimate the optical flow from timestamp t0 to timestamp t1, a spike stream should be collected containing (100 + dt) spike frames from t0, where dt represents the number of spike frames between t0 and t1. For instance, using a first-generation spike camera with a frequency of 20 kHz, when dt is set to 10, the necessary time to collect the spike stream for optical flow estimation would be approximately 5.5 ms. During inference, when utilizing a 3090 Ti GPU to estimate the optical flow between two timestamps, the computational time of our USFlow is 90.6 ms, whereas the computational time of the comparative method SCFlow is 236.9 ms.
>
> **Q2.Comparing the accuracy of the proposed method on standard rgb frames to show the benefits of using a high temporal resolution sensor for optical flow estimation.**\
> Typical RGB cameras usually operate at frequencies between 30Hz and 120Hz. The spike stream output by the spike camera takes advantage of its ultra-high temporal resolution, which can compensate for the deficiencies of low-frame-rate RGB frame sequences in optical flow estimation task. Taking a 120Hz RGB camera as an example, within a time span of (1/120)s, the RGB camera can only capture two RGB frames. Estimating nonlinear intricate motion between these two adjacent RGB frames by frame-based methods is often challenging. However, the spike stream, with its inherent advantage of extremely high temporal resolution, is capable of continuously recording dynamic processes. Thence, spike stream can be used to estimate nonlinear and non-uniform motion. Specifically, the (1/120)s can be divided into multiple short time spans. The motion within each short time span can be estimated on the corresponding spike stream. Eventually, by combining the motion information within these short time spans, the complete intricate motion within the (1/120)s can be estimated.
>
> **Q3.For people not familiar with the spike camera, it would be beneficial a more complete introduction to this type of sensor.**\
> In Section 3.1, we introduce the fundamental working mechanism of the spike camera. Some previously works, such as [a] and [b], provide a more comprehensive introduction to the spike camera. Thank you for your suggestions, we will cite these references containing detailed introductions of the spike camera.\
> [a] J. Zhao, et al. Reconstructing Clear Image for High-Speed Motion Scene With a Retina-Inspired Spike Camera. TCI 2022.\
> [b] T. Huang, et al. 1000x Faster Camera and Machine Vision with Ordinary Devices. Engineering 2022.

---

> > ### Comment · Reviewer_fVdD · 2023-08-16
> >
> > Thank you for your answers. I still believe that the points raised in the initial review are relevant and their discussion should be added to the article.

---

> > > ### Author Response · Authors · 2023-08-16
> > > **Thank you for your response**
> > >
> > > Thank you for your suggestions. The points you raised in the initial review are important, and we will add these discussions to the paper as you suggested.

---

### Author Response · Authors · 2023-08-21
**Thanks to reviewers**

We express our sincere gratitude to reviewers for their constructive suggestions. We will certainly add the corresponding discussions to this paper as reviewers suggested. Thank you for your approval of this work.

---

### Decision · Program_Chairs · 2023-09-21

**Decision:**

Accept (poster)

**Comment:**

The paper introduces a novel approach to unsupervised optical flow estimation using spike cameras. The methodology is well-explained, and the unique dataset provided could potentially be valuable for the research community. The paper was evaluated by five reviewers. Out of the five, four gave weak accept ratings, while one provided a borderline rating without expressing strong objections to accepting the paper. The authors are encouraged to consider the reviewers' feedback and incorporate the suggested revisions in the final version of the paper.